# Design and testing of multi-PMTs and Regression Studies for New Generation Water Cherenkov Detectors

Brenda E. Medina[1,a], Elías A. Avendaño[1,b], Rajesh R. Biswal[1,c], Karen S. Caballero-Mora[3,d], Edgar Chucuan[1,e], Eduardo De la Fuente[2,f], Giannina Dalle Mese[4,g], Luis E. Falcón-Morales[1,h], Christoper E. Falcón Anaya[1,i], Rodrigo Gamboa[1,j], Elrick E. Haces Gil[1,k], Rodrigo Medina[1,l], Felipe Orozco-Luna[2,m], Gilberto Rodriguez[1,n], Lazaro M. Salas[2,o], Rodrigo Salmón-Folgueras[1,p], Alejandro K. Tomatani-Sánchez[1,q] and Saul Cuen-Rochin[1,r]

1 Tecnologico de Monterrey, Escuela de Ingenieria y Ciencias.
2 Departamento y Licenciatura de Fisica, CUCEI, Universidad de Guadalajara.
3 Facultad de Ciencias en Física y Matemáticas, Universidad Autónoma de Chiapas.
4 Facultad de Ciencias de la Tierra y el Espacio, Universidad Autónoma de Sinaloa.

a elisa_medina@exatec.tec.mx   b a00833869@tec.mx   c rroshanb@tec.mx
d karen.mora@unach.mx   e edgar.chucuan@tec.mx   f eduardo.delafuente@academicos.udg.mx
g giannina@uas.edu.mx   h luis.eduardo.falcon@tec.mx   i christopher.falcon@tec.mx
j r.gamboa@tec.mx   k a00833699@tec.mx   l roymedina@exatec.tec.mx   m felipe.orozco@udg.mx
n a01635693@tec.mx   o lazaro.salas0881@alumnos.udg.mx   p jrsalmon@tec.mx
q katsumi@tec.mx   r saulcuen@tec.mx

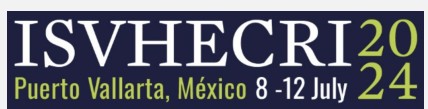

*22nd International Symposium on Very High Energy Cosmic Ray Interactions (ISVHECRI 2024) Puerto Vallarta, Mexico, 8-12 July 2024*

## Abstract

**This paper presents contributions to the mechanical-electronic design and testing of a multi-PMT (mPMT) vessel for Hyper-Kamiokande (HK) experiment. Computational results, alongside physical testing, demonstrate satisfactory performance while highlighting potential areas for improvement. Additionally, we conduct regression studies aimed at reconstructing kinematic variables; position, direction and energy, for the Intermediate Water Cherenkov Detector (IWCD), utilizing ResNet-50 for analyzing neutrino beam samples. These findings help to improve neutrino detection methodologies.**

## 1   Introduction

Hyper-Kamiokande (HK) [1] is a prominent international particle physics project focusing in detecting neutrinos and studying fundamental particle properties[1].  Within HK, the development and testing of mPMTs [2] is crucial to ensure proper functioning under associated ultra-pure water and an hydrostatic pressure conditions caused by 70 m of water.  This proceeding

---

[1]To know more refer to the proceedings *Hyper-Kamiokande* by S. Cuen-Rochin.

6  focuses in the improvement of the structural design, testing and fabrication of a mPMT, to-
7  gether with a regression analysis to reconstruct kinematical variables for Intermedi- ate Water
8  Cherenkov Detector (IWCD), a subdetector from the HK observatory complex.

## 2   IWCD Regression Studies

10  Systematic uncertainty reduction in HK demands old and new detectors to be repurposed and
11  constructed, respectively [3] [4]. IWCD is a novel cylindrical water tank detector (6 m in length
12  and 8 m in width), capable of vertically moving through a pit to measure neutrino fluxes at
13  different off-axis angles. The detector instrumented with mMPTs, can contain muons up to
14  1 GeV/c, providing a comprehensive measurement capability [5]. This analysis uses Particle
15  Gun (PG) data and the Neutrino Beam Sample (NBS) datasets. PG data is generated using
16  WCSim[2], simulating simple interactions where some of the associated kinematic variables are
17  uniformly distributed. In contrast, NBS data shows complex distributions, better reflecting
18  real-world conditions.
19      Multiple ResNet-50 models were trained on PG data for electron and muon events, with
20  separate models for each kinematic variable; position, direction and energy, and particle type
21  $\mu$ and $e^-$. The performance of these models was then evaluated using the NBS dataset. Events
22  were selected to evaluate model performance, where the energy is less than 1 GeV above the
23  Cherenkov threshold, a minimum of 25 hits, and a minimum distance of 50 cm, between the
24  vertex and the detector wall. A fiTQun[3] flag was also applied to indicate whether the particles
25  were fully contained in the detector. This event selection matches the range of values in the
26  PG dataset, as the models were trained within this range, necessitating the filtering of events
27  that meet these criteria. Additionally, the model's performance was evaluated across various
28  charged-current interaction channels, including Charged-Current Quasi-Elastic (CCQE), thee
29  purely quasielastic cross section (1p1h), the partile emission channel (2p2h), and Single Pion
30  Production (SPP, CC1$\pi$).
31      After evaluating the model with the NBS dataset, comparison performance is attained via
32  residual histograms (figure 1). The model consistently achieves better reconstruction for PG
33  data compared to NBS data across all variables, nonetheless for the golden channel category
34  (1p1h), the model shows the best performance among the different charged-current interac-
35  tion types, in the NBS dataset. Predictions for the 2p2h category are reasonably accurate,
36  while CC1$\pi$ events are consistently predicted poorly across all variables.
37      A plausible explanation for this behavior is that CCQE events are simpler, involving fewer
38  particles, whereas CC1$\pi$ events typically produce multiple rings, increasing the complexity of
39  the event, making accurate reconstruction more difficult for the model.

## 3   Mechanical Design, Simulation and Testing Results

41  Stress analysis, including buckling, stress, and deformation simulations, was performed in
42  SolidWorks using a 3D model of the mPMT (Figure 2 (a)) to evaluate the effects of various
43  stresses and materials on its main components (Figure 2 (b)) and identify potential improve-
44  ments in the mechanical design.
45      A key simulation, the stress test, revealed areas where the device experiences high stress
46  due to hydrostatic pressure, as indicated by the color scale in Figure 3.

---

[2]https://github.com/WCSim/WCSim
[3]Legacy software used in Super-Kamiokande

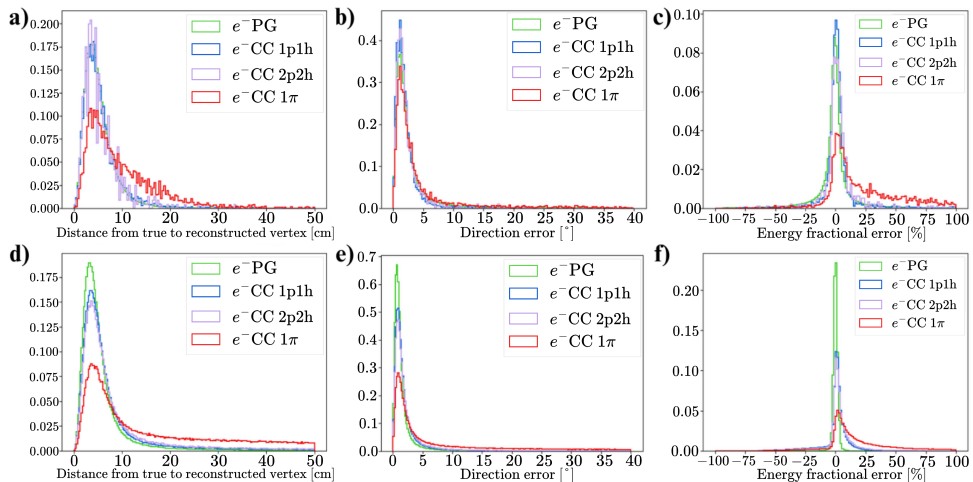

Figure 1: Reconstruction error residuals for $e^-$ (top row), and $\mu$ (bottom row) events using PG and NBS, for distance (left), direction (center) and energy (right). Copyright our own.

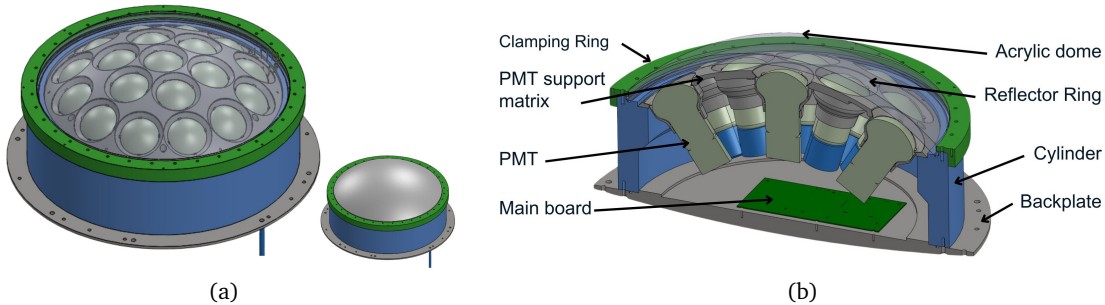

Figure 2: (a) mPMT SolidWorks 3D model. (b) Schematic of the mPMT. Copyright for figures our own, and original design by Hyper-Kamiokande collaboration.

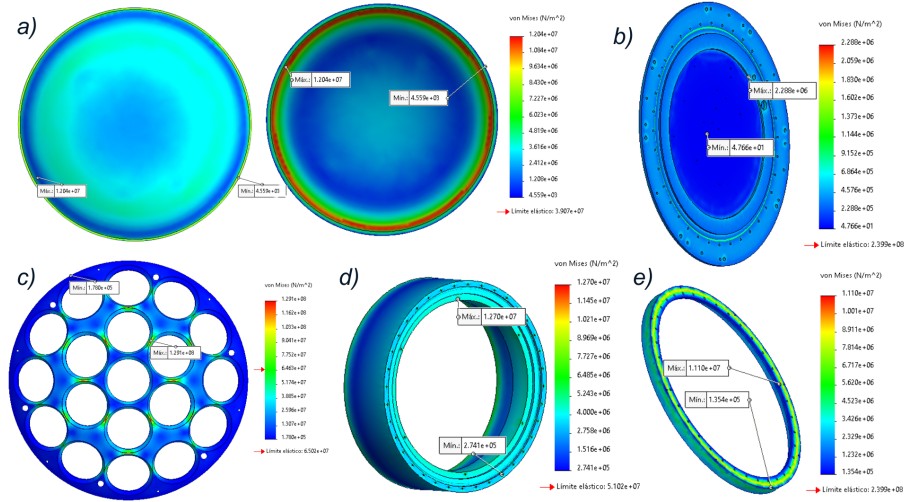

Figure 3: Stress tests results. Copyright for figure our own, and original design by Hyper-Kamiokande collaboration. a) Top and bottom of Plaskolite General Purpose Acrylic Resin Dome. b) Backplate of AISI-316 material. c) PMT vessel of ABS material. d) Cylinder of POM-C material. e) Clamping Ring of AISI-316 material.

The components most affected were the Acrylic Dome and the mPMT vessel. The bottom edge of the Acrylic Dome, which has an elastic limit of $3.907 \times 10^7\ N/m^2$, experienced a maximum stress of $1.204 \times 10^7\ N/m^2$. While this stress is significant, it remains below the material's elastic limit, ensuring that the dome will not deform under these conditions. In contrast, the mPMT vessel experienced a maximum stress of $1.291 \times 10^8\ N/m^2$ at the joints, exceeding the ABS material's elastic limit of $6.502 \times 10^7\ N/m^2$. Although the mPMT vessel is not directly exposed to hydrostatic pressure, the joints are vulnerable and may require reinforcement to prevent structural failure.

To address these weaknesses, additional computational simulations were conducted for each main component using nine alternative materials to identify more resistant options. Table 1 summarizes the materials that best withstand hydrostatic pressure based on the simulation results.

Table 1: Results of best materials for each component.

| Component | Original material | Best material | | |
|---|---|---|---|---|
| | | Buckling test | Stress test | Deformation test |
| Dome | Plexiglas GS-UVT | General Purpose Acrylic Resin | Röhm ACRYLITE | Both |
| PMT vessel | ABS, PET-G and resins | ABS (ABS834G40L) | PET-G | |
| Cylinder | POM-C | Stainless steel AISI-304 | | |
| Backplate | AISI-304 | Stainless steel AISI-316 | | |
| Clamping Ring | | | | |

## 3.1 Metrology and testing results of mPMT base design

A metrology study of the mPMT vessel components was conducted to verify dimensional accuracy for the precise 3D printing of the vessel's PMT support matrix. Physical measurements of the prototype were compared to the SolidWorks model, with a margin of error below 1% [6]. Among the 416 measurements collected, most absolute errors between the measured and reference dimensions were within tolerance, validating the 3D printing of the PMT support matrix. Furthermore, the Mexican collaboration designed and constructed mPMT support structures prototypes for the barrel, top, and bottom configurations of Hyper-K's main tank Inner Detector. Such bases were tested by simulating earthquake frequencies demonstrated the structural stability of the mPMT vessel in all configurations—top, bottom, and barrel—within the water tank, as shown in Figure 4. Additionally, packaging, transportation, and installation tests were performed to evaluate the durability of the packaging designs under various conditions. Vibration, compression, and humidity tests confirmed the prototypes' packaging resistance to transportation stresses, including handling frequencies, stowage loads, and environmental humidity.

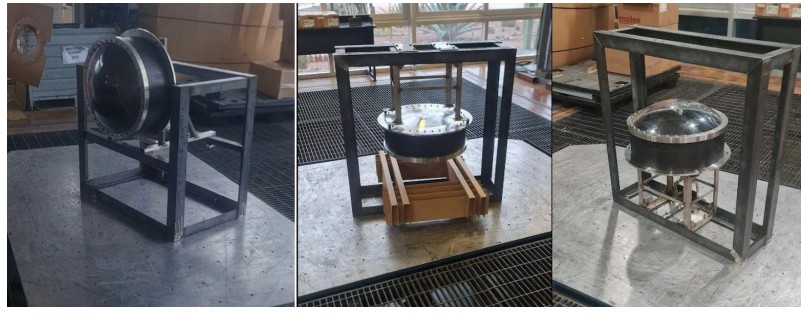

Figure 4: mPMT and support structures vibration test for the top, bottom and barrel positions. Copyright our own.

## 3.2  Reflector rings results

A prototype of aluminum Reflector Rings was designed and manufactured to evaluate an alternative of fabrication method at the facilities of Tecnologico de Monterrey. Six 2D prototypes of Reflector Rings were cut from a 25-gauge (0.46$mm$ thick) aluminum sheet using a plasma cutter. The dimensions and angles were close to the design specifications. However, defects were observed on the edge of the rings, which could be attributed to factors related to the cutting process[4].

# 4  Photonic Characterization of Aluminum Rings and PMTs

Within the HK project, characterization of the PMTs and its reflector rings is essential. This process involves configuring two fundamental systems: the electrical setup for PMT operation and the optical setup for controlled emission of individual photons. By counting photons with and without the rings, we can measure their added efficiency.

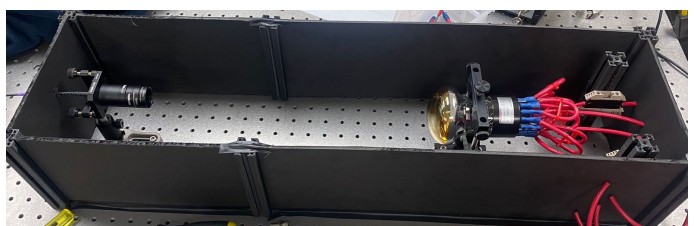 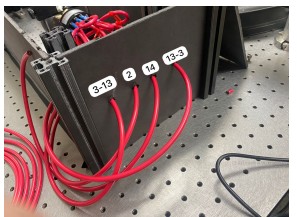

Figure 5: Experimental setup: (left) Black Box for Single Photon Emission with Weak Coherent Source and PMT; (right) Electrical Pin-Outs. Copyright our own.

The aluminum rings have shown to increase the PMT efficiency by 20%. Using COMSOL raytracing simulations, this efficiency increase was computed at 15.8% for a single PMT, and 18.6% for the mPMT assembly. In the experimental assembly described further, we aim to confirm the 15.8% increase in efficiency by counting photons experimentally.

## 4.1  Electrical Circuit

The circuit required includes a voltage division system, which allows for proper voltage distribution across various PMT pins. When a photon is detected by the PMT, it will decrease its internal resistance allowing the cascade of electrons. This causes a voltage drop and a current spike (only detectable by oscilloscopes). These fluctuations in voltage translate into the photon count [7]. When the power supply is able provide more current, the voltage change returns to baseline faster, translating into higher time resolution when detecting photons. Two voltage distributions were used, one specified by Hamamatsu for models R14374 and R14689 datasheets, and another one specified by the Hyper-Kamiokande Collaboration. Experimental setup shown in Figure 5.

## 4.2  Single Photon Emission Optical Setup

For photon emission the "weak coherent source" (WCS) technique was employed. A 402 nm laser simulating Cherenkov radiation, is attenuated using neutral density filters to achieve individual photon-level emission. The laser source produces coherent states:

---

[4]Such as the parameters or maintenance conditions of the plasma cutter used.

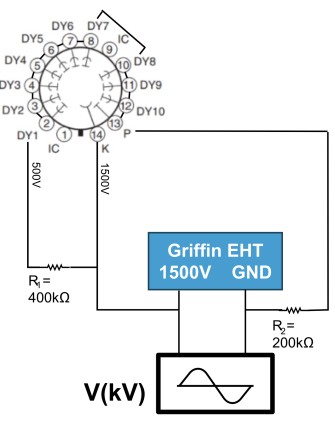

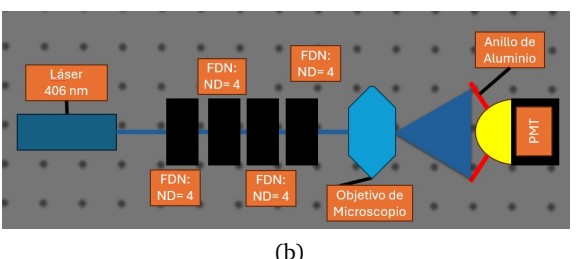

(a)

(b)

Figure 6: (a) Voltage divider and supply suggested by Hamamatsu for PMT R14374. Copyright our own. (b) Proposed experimental setup. Copyright our own.

104
$$|\alpha\rangle = e^{-\frac{|\alpha|^2}{2}} \sum_{n=0}^{\infty} \frac{\alpha^n}{\sqrt{n!}} |n\rangle \qquad |\alpha\rangle \xrightarrow{\text{BS}} |\eta\alpha\rangle \qquad ND = -\log_{10}\left(\frac{3.105 \text{ eV s}^{-1}}{1 \text{ mW}}\right)$$
$$\approx 15$$

105  Our goal is to reduce to a single coherent state with an average of one photon, achieved
106  by setting $|\alpha|^2 = 1$. Applying attenuators is equivalent to using beam splitter operators, where
107  $\eta$ is the complex transmissivity coefficient. We adjust $\eta$ such that $|\eta|^2|\alpha|^2 = 1$, resulting in
108  a coherent state with an average photon number of 1. The attenuation calculation uses the
109  power of a single photon per second and the laser power, as shown in the ND equation above.

### 110  4.3  Proposed Experimental Setup

111  Voltage divider and supply suggested by Hamamatsu for PMT R14374 in Figure 6 (a). By
112  having quality measurements of Photon Counts, we could use a conical release pattern (Figure
113  6 (b)) with a Microscope Objective to calculate the added efficiency provided by the aluminum
114  ring. In order to verify the 20% efficiency increase reported in Hyper-K internal technical
115  reports, and corroborate the 15.8% improvement predicted by COMSOL simulations.

## 116  5  Conclusion

117  Preliminary results for the mPMT vessel and top, bottom, and barrel support structures were
118  satisfactory, establishing a solid foundation for future studies, ensuring its structural resis-
119  tance to hydrostatic pressure via the proposed improvements. Regarding the IWCD regression
120  studies, results demonstrates the viability of employing deep learning models, particatarly
121  ResNet-50, to reconstruct muon and electron e.vents in the IWCD detector. While the ResNet-
122  50 models show promise, particularly in analyzing distinct interaction types like CCQE, future
123  work should aim at improving performance for events with higher energy. Next steps will in-
124  clude single photo detection characterization on Hamamatsu's R14374 PMT, and the also for
125  the mPMT.

## Acknowledgements

The authors would like to thank CONAHCyT projects CF-2023-G-643 and CBF2023-2024-427, for supporting this work.

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
