# Peer review of "Design and testing of multi-PMTs and Regression Studies for New Generation Water Cherenkov Detectors"

_SciPost Physics Proceedings_

## Round 1 · Referee Report · Anonymous (Referee 1) · 2025-5-21

Strengths

1) Good overview on a simulation study on reconstruction accuracy using a modern Convolutional Neural Network. 2) very broad overview over ange of different aspects connected to the mPMT module from Hyper Kamiokande, and the IWC Water Cherenkov detector.

Weaknesses

1) The paper stays rather superficial on the presented topics. 2) Description of the experimental setup (Chapter 4) is rather poor.

Report

review of paper
Design and testing of multi-PMTs and Regression Studies for New Generation Water Cherenkov Detectors
by B. E. Medina et al

brief summary:
The paper gives a very brief summary overview on various contributions regarding the multi-PMT vessel for the Hyper Kamiokande experiment.
Studies on the reconstruction accuracy achieved by applying "ResNet-50" CNNs for reconstruction of various kinematic variables for the Intermediate Water Cherenkov Detector are shown (which is a subdetector of the Hyper Kamiokande setup).
FEM stress analysis using SolidWorks of various components of the multi-PMT module under large hydrostatic pressure are discussed, and an experimental setup is presented for future measurements of the photon gain of an aluminum reflector ring (similar to a Winston cone) using an attenuated laser as single photon source.

paper evaluation:
Being a proceeding, the paper covers - without going into too much depth - several fairly distinct aspects on the simulation analysis of the IWC Detector, as well as construction aspects and FEM siimulations of the mPMT module, and an experimental setup for measurements on photon efficiency. Not much effort is spent to interconnect these three distinct topics.
Chapter 2 ("IWCD Regression studies"), covering the simulation studies on reconstruction performance of the IWC detector, is the most informative. This chapter is fairly clearly written and structured.
Chapter 3 ("Mechanical design, simulation and testing") is presenting the FEM studies of various mPMT components under water, comparing simulated von-Mises stress results to the elasic limits comapring different materials. Further aspects covered are the accuracy of 3d-printed parts for the mPMT module, and fabrication of the reflector ring. Also this chapter is fairly well written and structured.

Chapter 4, presenting the measurement setup for photon yield of the reflector rings, needs some more work before publication: The description of the working principle of the PMT divider circuit in chapter 4.1 should be shortened, and is also misleading and partly wrong (see detailed comments below). Figure 6.a does not make any sense.
The formulas in line 104, and the text in line 103-109 are a complicated way of basically saying that a laser light source is attenuated to the single photon level, and that the measured photon yield is used to evaluate the performance of the reflector ring.
Chapter 4.3 "Proposed experimental setup" is hardly understandable, contains bad grammar (missing verbs), and needs to be rewritten.

Chapter 5 is concluding with a concise summary, and apart from a few typos well written.
In general, the paper would certainly profit careful from proof reading by a native speaker. Given the fact, that this is a proceeding and not a full scientific paper i would recommend it for publication but only after some modifications as detailed below.

Requested changes

detailed comments, corrections, typos, ...:

Abstract: "[...] testing of a multi-PMT (mPMT) vessel for THE Hyper-Kamiokande (HK) experiment."

line 2: focusing ON detecting neutrinos...

line 4: what is ment with "under associated ultra-pur water" ? i suggest "to ensure proper functioning in ultra pure water conditions at a hydrostatic pressure corresponding to a depth of 70m in water."

line 6: focuses ON the improvement

line 7: "to reconstruct kinematical variables using the Intermediate Water Cherenkov Detector (IWCD)"

line 13: "The detector, instrumented with mPMTs (not mMPTs...), can restrain muons up to 1 GeV/c,..:"

line 19: ResNet-50 is a term not known to every reader. I suggest to add a sentence explaining what ResNet50 is about.

line 25: "The range in kinematic variables covered by this event selection matches well the PG dataset used for training of the models." Skip "necessitating the filtering ..."

line 28: "the (not thee) purely quasielastic cross section."

line 31: "After evaluating the model using the NBS dataset, a performance comparison is obtained based on residual histograms..."

line 32: "The model consistently achieves better reconstruction performance for the PG data set as compared to the NBS data. Among the different CC-interaction channels, the model performs best for the golden channel category (1p1h)."

line 40: "Mechanical Design, Simulation and Testing of the mPMT module"

line 46: "hydrostatic pressure (70m of water-equivalen)"

Figure 1 caption: in all captions, you state "Copyright our own". I guess, this comment should be removed for the final submission (in all captions) !?

Figure 3 caption: "Von Mises stress results for the different objects, for a simulated water depth of 70~m".

line 50: "the dome will not deform": Actually it will deform under pressure, but elastically ! You just dont cross the elastic limit.

Table 1: I would put the caption below the table, as done for all the figures as well...

line 66: "support structure prototypes" (not "structures")

line 67: "demonstrated" -> "demonstrating"

line 74: "Reflector ring results"

line 75: "A prototype of THE aluminum Reflector Rings" From the article, it is not clear, what actually these aluminum Reflector rings do. Can you add one explanatory sentence at beginning of this sub chapter ?

line 76: "to evaluate an alternative fabrication method" (skip "of")

line 85: "we aim to determine the efficiency gain by adding these reflector rings."

line 88: "With help of the experimental setup (as shown in Fig. 5, more details in the next paragraphs) we aim to confirm the 15.8% increase in efficiency due to the reflector ring, by explicitly counting the number of detected photons experimentally.

line 90 ff: The description of the working principle of PMT and divider circuit is missleading and partly wrong. I also suggest to shorten it, since working princple of a PMT does not need to be explained in this detail. Also: Timing precision will improve with increasing HV supply voltage. The divider current (and not the current capability of the HV supply, this must be simply sufficient to drive the voltage divider) will determine the rate stability of the PMT. This has nothing to do with the timing precision for an individual pulse !

I suggest: "The Photomultiplier is connected to a Voltage Divder, providing the individual dynode voltages from a common HV supply. Every single incoming photon is converted to a photo-electron, amplified by the gain of the PMT, and generating a charge pulse on the output. The number of detected photons can thus be determined by counting these output pulses. Two types of Voltage Dividers were used: the first one specified by Hamamatsu for usage with PMTs R14374 and R14689, the second one specified by the HK collaboration."

line 90: The caption reads "Electrical Circuit". I would suggest to change it to "Electrical PMT Readout", and in the text add few lines how the PMT signals are actually processed. Are they discriminated ? And counted using a digitizer ? Or counted using a scope ??? At the moment, the subchapter only covers the PMT voltage divider. In this case, i would change the caption to "PMT Voltage Divider" or "PMT signal extraction".

line 103: "to achieve single-photon level".

Figure 6a: Figure a is misleading. The Voltage divider is missing. Certainly there is not only a single voltage (500V) connected to Dynode 1 ! What is V(kV) ? Should this symbolize the signal measurement ? But the signal would be measured at the Anode (P) Why is the Anode (P) connected to GND ? This figure needs quite some modification.

Figure 6b: What is the grey triangle ? Is this your "conical release pattern" mentioned in the text ?

line 104: I suggest to skip the formula, and also the state transition. Or add some more explanation, what the formula means. In the end, you simply illuminate with single photons after attenuation of the laser, and count the number of detected photons. If you stick with the formulas : What is alpha, what is eta ? what is BS ?

line 111: There is a verb missing in this sentence. What do you want to say here ?

line 112: What do you mean with "By having quality measurements of Photon counts" ??? What is "quality measurements" ?

line 112: What is a "conical release pattern" ?

line 114: "In order to verify... and corroborate the improvement" - again, there is verb missing. This is not a full sentence !

line 110-115: Actually, this whole subchapter 4.3 is more like a keyword collection, and needs reformulation. It should (!) give an overview of your experimental setup and proposed measurements. Right now it is hard to understand.

line 120: "results demonstrated"

line 120: "particatarly" -> "particularly"

line 121: "and electron events" (skip ".")

line 122: "While the ResNet50 models show promising results"

line 124/125: "and the also for the mPMT" ????

reference [1]: Why not mention only one author here + et. al: H.-K. Proto-Collaboration: K. Abe et al, "Hyper-Kamiokande design report",

Recommendation

Ask for major revision

---

## Editorial Decision

accepted_in_target_journal